# Incidence of major depressive disorder over time in patients with liver cirrhosis: A nationwide population-based study in Korea

**Gi Hyeon Seo**[1], **Jeong-Ju Yoo**[2]*

**1** Health Insurance Review and Assessment Service, Wonju, South Korea, **2** Department of Gastroenterology and Hepatology, Soonchunhyang University School of Medicine, Bucheon, South Korea

* puby17@naver.com

**Data Availability Statement:** The access to raw data of the Korean Health Insurance Review and Assessment (HIRA) service is regulated by the Rules for Data Exploration and Utilization of the HIRA. Researchers can use the confidential data

## Abstract

There is yet to be a large-scale longitudinal study on the course of depression incidence within the duration of cirrhosis. The aim of this study is to analyze the incidence of depression from before to after diagnosis of cirrhosis over time. Incidence Rate (IR) was defined as the number of newly diagnosed patients with MDD divided by the sum of observation periods by using claims database in Korea. Incidence Rate Ratio (IRR) was defined as the IR in the specific interest period divided by the IR in the control period. The control period was defied as 1 to 2 years before diagnosis of cirrhosis. The IRs before and after cirrhosis diagnosis were 3.56 and 7.54 per 100 person-year, respectively. The IRR was 2.12 (95% confidence Interval: 2.06–2.18). The IRR of developing depression mildly increased before diagnosis of cirrhosis (-360 days to -181 days, IRR 1.14, p < 0.001; -180 days to -90 days, IRR 1.24, p < 0.001; -90 days to -31 days, IRR 1.56, p < 0.001) and rapidly increased immediately after diagnosis of cirrhosis (+30 days to +89 days, IRR 2.12, 95% confidence interval: 2.06–2.18, p < 0.001). The pattern of increasing depression immediately after the diagnosis of cirrhosis was observed equally in all sexes and ages. Thus, clinicians must pay close attention to screening for depression within the first three months of liver cirrhosis diagnosis.

## Introduction

Liver cirrhosis is a noteworthy subject of research in gastroenterology, characteristic in its irreversibility and progression of complications such as ascites and hepatic encephalopathy. Such patients requiring endless medical attention face a high prevalence of depression, a bleak trend evident among chronic conditions [1]. The course of depression varies among conditions: in patients with cancer or chronic lung disease, the incidence of depression increases immediately after diagnosis and decreases over time, whereas in patients with heart disease or arthritis, the incidence of depression increases not only immediately after diagnosis but also over time [2–4]. However, studies on the incidence of depression in patients with cirrhosis are extremely rare, with most pre-existing studies limited in long-term analysis due to their cross-sectional design [5, 6]. To the best of our knowledge, there is yet to be a large-scale longitudinal study on the course of depression incidence within the duration of cirrhosis. Therefore, using a

from the HIRA only after receiving an approval for the data use from the data access committee of the HIRA (Big Data Division, Healthcare Data Convergence Department, Korean Health Insurance Review and Assessment Services, opendata@hira.or.kr). Since all data are delivered anonymously, none of the researchers can know or access to any potentially identifying personal information of subjects, such as names, addresses, date of birth, and etc.

**Funding:** This study was supported by the Soonchunhyang University Research Fund. The funders had no role in study design, data collection and analysis, decision to publish, or preparation of the manuscript.

**Competing interests:** The authors have declared that no competing interests exist.

population-based cohort from South Korea, we analyzed the increase of depression incidence from before to after diagnosis and the course of depression incidence over time after diagnosis of cirrhosis.

## Materials and methods

Our study utilized the claims database from the Health Insurance Review and Assessment Service (HIRA), which is generated when healthcare providers submit claims to HIRA for reimbursement reviews. Comprised of information on nearly 99% of the resident population, HIRA is undeniably one of the most credible resources for medical research in South Korea.

Out of 787,505 adult patients treated for liver cirrhosis from 2010 to 2018, patients groups including those aged 18 years or younger, or 89 years or older, considering the average life expectancy in Korea (n = 7,953); those diagnosed with cirrhosis from 2007 to 2009, to include only newly diagnosed cirrhosis patients during the study period; (n = 175,608); those diagnosed with depression between January 2007 and 720 days prior to the diagnosis of cirrhosis to include only newly diagnosed MDD [7] (n = 93,164) and those with an incorrect mortality code such as the date of death precedes the date of diagnosis due to administrative error; (n = 43) were excluded (Fig 1). Finally, a total of 510,737 cirrhotic patients were enrolled in the analysis. The study protocol was approved by HIRA's the Institutional Review Board (IRB Number 2022–045) and conformed to the ethical guidelines of the World Medical Association Declaration of Helsinki. Liver cirrhosis and major depressive disorder were defined based on the ICD-10 code: i) liver cirrhosis (K70.2, K70.3, K74.x), ii) major depressive disorder (F32.x). First-ever documentation of cirrhosis was defined as a new patient with no treatment for liver cirrhosis for at least 3 years (January 2007 –December 2009 wash-out), and major depressive disorder (MDD) was defined a patient who had no treatment for MDD for at least 1 year (between January 2007 and 720 days prior to the diagnosis of cirrhosis) as a new patient.

The time interval was determined considering on the length of depressive episode, average duration of evaluation of improvement in depression, and the impact of acute event such as diagnosis of liver cirrhosis [8–10]. Considering that cirrhosis and MDD are chronic diseases, the baseline was set to 1 year 2 years ago before diagnosis. Assuming that the patient's condition can change more frequently around the onset of cirrhosis, the analysis was conducted at shorter intervals of 180 days, 90 days, and 60 days closer to the date of diagnosis of cirrhosis. Based on this information, the duration of liver cirrhosis was divided into a total of eight periods: before diagnosis of cirrhosis (from day -720 to -361 (reference period), day -360 to -181, day -180 to -91, day -90 to -31), after diagnosis of cirrhosis (from day 30 to 89, day 90 to 179, day 180 to 359, day 360 to 719).

The primary outcome of the study was the extent of major depressive disorder incidence increase from before to after diagnosis of cirrhosis. Incidence rate (IR) was defined as the number of newly diagnosed patients with MDD divided by the sum of observation periods. Incidence rate ratio (IRR) was defined as the IR in the specific interest period divided by the IR in the control period. The 95% confidence interval (CI) and p-value were calculated using a Poisson approximation. In addition, we conducted stratified analysis according to age and sex considering the past results that the prevalence of depression differs according to age and sex [11–16].

## Results

Out of 510,737 cirrhotic patients, 349,098 (68.4%) were male, and the mean age was 55.8 ± 13.5 years. The IR of MDD at -720 days to -361 days before the diagnosis of cirrhosis was 3.56 persons per 100 person-year, and the IR of MDD at -360 days to -181 days, -180 days

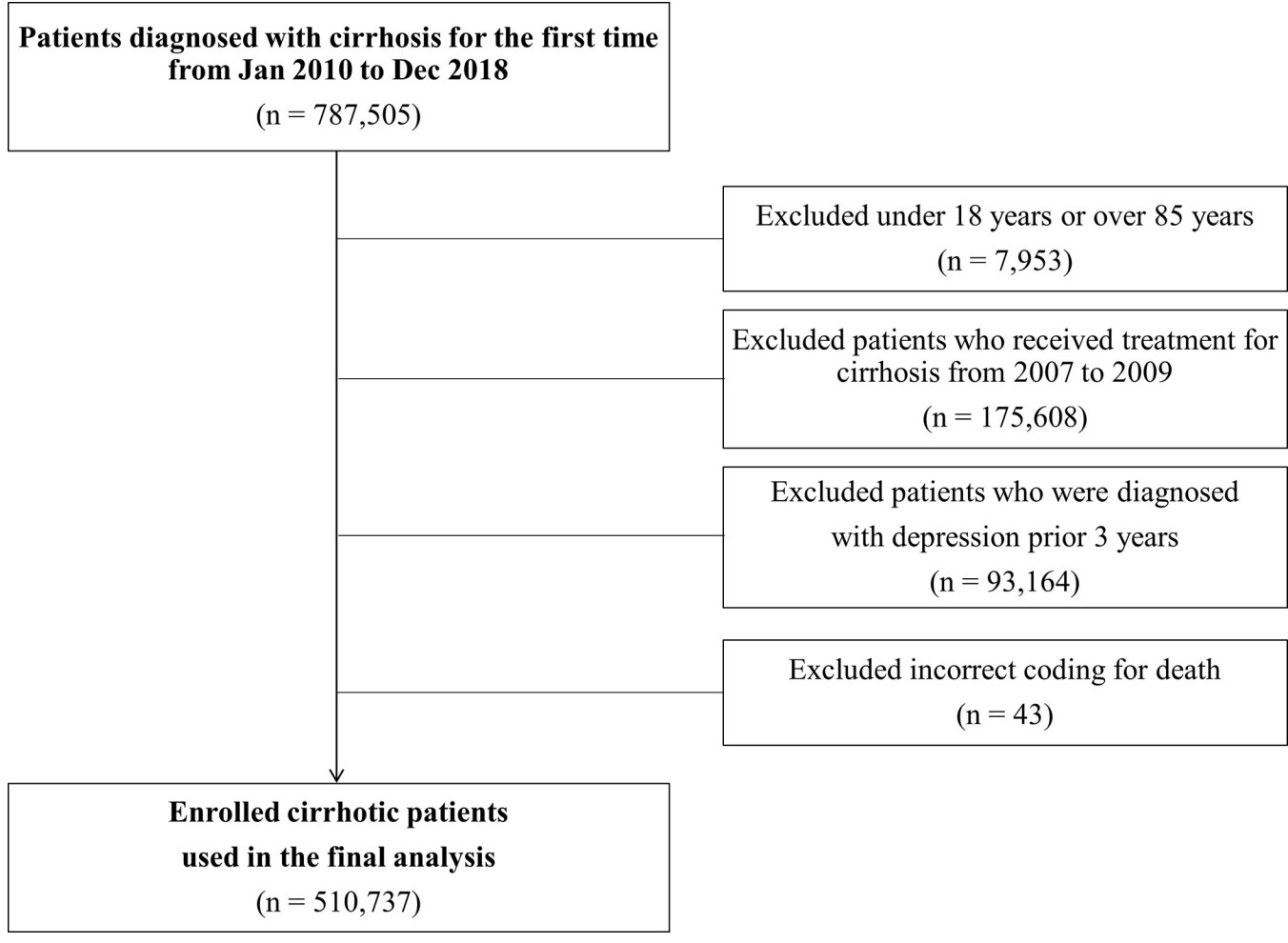

**Fig 1.**

to -91 days, and -90 days to -31 days was 4.04, 4.41, and 5,56, respectively. The IR after diagnosis of cirrhosis was the highest at +30 days to +89 days at 7.54 persons per 100 person-year, and then at +90 days to +179 days, +180 days to +359 days, and +360 days to +719 days, with 5.82, 4.99, and 4.36, respectively. Although IR decreased slightly compared to immediately after diagnosis, it was still higher than the baseline (Table 1, Fig 2A).

The IRR of developing depression mildly increased before diagnosis of cirrhosis (-360 to -181, IRR 1.14, p < 0.001; -180 to -90, IRR 1.24, p < 0.001; -90 to -31, IRR 1.56, p < 0.001) and rapidly increased immediately after diagnosis of cirrhosis (+30 to +89, IRR 2.12, 95% confidence Interval: 2.06–2.18, p < 0.001). After that, the rate of increase in IRR decreased over time (+90 to +179, IRR 1.64; +180 to +359, IRR 1.40, p < 0.001; +360 to +719, IRR 1.22, p < 0.001), compared to the rate of increase during the period of reference (from day -720 to -361) (Fig 2A). Our study further conducted stratified analysis according to sex and age. The pattern of increasing depression immediately after the diagnosis of cirrhosis was observed equally in all sexes and ages. The IR of depression within three months after diagnosis of cirrhosis was 5.05, 6.69, 4.48, 6.55 per 100 person-year in men, women, young patients and elderly patients, respectively, with the IRR of 1.62, 1.48, 1.55, and 1.58, respectively (Fig 2B and 2C).

**Table 1. Incidence of depression in patients with cirrhosis.**

| Day from diagnosis (days) | Pre-diagnosis period (-720)-(-361) | Pre-diagnosis period (-360)-(-181) | Pre-diagnosis period (-180)-(-91) | Pre-diagnosis period (-90)-(-31) | Post-diagnosis period (+30)-(+89) | Post-diagnosis period (+90)-(+179) | Post-diagnosis period (+180)-(+359) | Post-diagnosis period (+360)-(+719) |
|---|---|---|---|---|---|---|---|---|
| **Total** | | | | | | | | |
| All patients | 510,737 | 492,882 | 483,027 | 477,724 | 447,148 | 428,092 | 412,123 | 390,503 |
| person-year | 501,962 | 244,017 | 120,116 | 79,286 | 72,768 | 6,102 | 9,999 | 6,303 |
| depression case | 17,855 | 9,855 | 5,303 | 4,407 | 5,487 | 104,876 | 200,373 | 374,221 |
| Incidence rate (per 100 person-year) | 3.56 (3.51–3.61) | 4.04 (3.96–4.12) | 4.41 (4.30–4.54) | 5.56 (5.40–5.72) | 7.54 (7.34–7.74) | 5.82 (5.67–5.97) | 4.99 (4.89–5.09) | 4.36 (4.29–4.42) |
| Incidence rate ratio (IRR) | 1 (ref) | 1.14 (1.11–1.16) | 1.24 (1.21–1.28) | 1.56 (1.52–1.61) | 2.12 (2.06–2.18) | 1.64 (1.59–1.68) | 1.40 (1.37–1.43) | 1.22 (1.21–1.24) |
| P-value (IRR) | | < 0.001 | < 0.001 | < 0.001 | < 0.001 | < 0.001 | < 0.001 | < 0.001 |
| **Stratification by sex** | | | | | | | | |
| Incidence rate (per 100 person-year) | | | | | | | | |
| Male | 3.11 (3.05–3.17) | 3.64 (3.55–3.73) | 3.98 (3.85–4.12) | 5.05 (4.86–5.24) | 7.51 (7.27–7.76) | 5.84 (5.66–6.02) | 4.94 (4.82–5.06) | 4.23 (4.15–4.31) |
| Female | 4.53 (4.42–4.63) | 4.92 (4.76–5.08) | 5.37 (5.14–5.61) | 6.69 (6.37–7.02) | 7.60 (7.25–7.97) | 5.78 (5.53–6.04) | 5.09 (4.92–5.27) | 4.62 (4.50–4.75) |
| Incidence rate ratio (IRR) | | | | | | | | |
| Male | 1 (ref) | 1.17 (1.14–1.20) | 1.28 (1.24–1.33) | 1.62 (1.56–1.69) | 2.41 (2.34–2.50) | 1.88 (1.82–1.94) | 1.59 (1.55–1.63) | 1.36 (1.33–1.39) |
| P-value (IRR) | | < 0.001 | < 0.001 | < 0.001 | < 0.001 | < 0.001 | < 0.001 | < 0.001 |
| Female | 1 (ref) | 1.09 (1.05–1.12) | 1.19 (1.13–1.24) | 1.48 (1.40–1.55) | 1.68 (1.60–1.76) | 1.28 (1.22–1.34) | 1.13 (1.09–1.17) | 1.02 (0.99–1.05) |
| P-value (IRR) | | < 0.001 | < 0.001 | < 0.001 | < 0.001 | < 0.001 | < 0.001 | 0.121 |
| **Stratification by age** | | | | | | | | |
| Incidence rate (per 100 person-year) | | | | | | | | |
| Young (<55) | 2.89 (2.82–2.96) | 3.26 (3.16–3.37) | 3.65 (3.49–3.81) | 4.48 (4.27–4.70) | 6.63 (6.37–6.90) | 5.04 (4.85–5.23) | 4.24 (4.11–4.37) | 3.59 (3.51–3.68) |
| Old (> = 55) | 4.16 (4.08–4.24) | 4.75 (4.63–4.87) | 5.12 (4.95–5.30) | 6.55 (6.31–6.81) | 8.41 (8.12–8.71) | 6.59 (6.37–6.81) | 5.75 (5.60–5.90) | 5.16 (5.05–5.26) |
| Incidence rate ratio (IRR) | | | | | | | | |
| Young (<55) | 1 (ref) | 1.13 (1.09–1.17) | 1.26 (1.21–1.32) | 1.55 (1.48–1.63) | 2.29 (2.20–2.39) | 1.74 (1.67–1.81) | 1.47 (1.42–1.51) | 1.24 (1.21–1.27) |
| P-value (IRR) | | < 0.001 | < 0.001 | < 0.001 | < 0.001 | < 0.001 | < 0.001 | < 0.001 |
| Old (> = 55) | 1 (ref) | 1.14 (1.11–1.17) | 1.23 (1.19–1.28) | 1.58 (1.51–1.64) | 2.02 (1.95–2.10) | 1.58 (1.53–1.64) | 1.38 (1.34–1.42) | 1.24(1.21–1.27) |
| P-value (IRR) | | < 0.001 | < 0.001 | < 0.001 | < 0.001 | < 0.001 | < 0.001 | < 0.001 |

## Discussion

Our study demonstrated that the incidence of depression increases immediately after diagnosis of cirrhosis and gradually decreases thereafter, regardless of sex and age of involved subjects.

The prevalence of MDD according to the course of chronic disease shows a different pattern for each chronic disease [17, 18]. In heart disease and arthritis, depression tends to increase over time, whereas in cancer and chronic lung disease, MDD increases immediately after diagnosis and decreases over time. In our study, MDD in cirrhosis was also observed in a pattern similar to that of cancer and chronic lung disease. In our study, MDD in liver cirrhosis

**A**

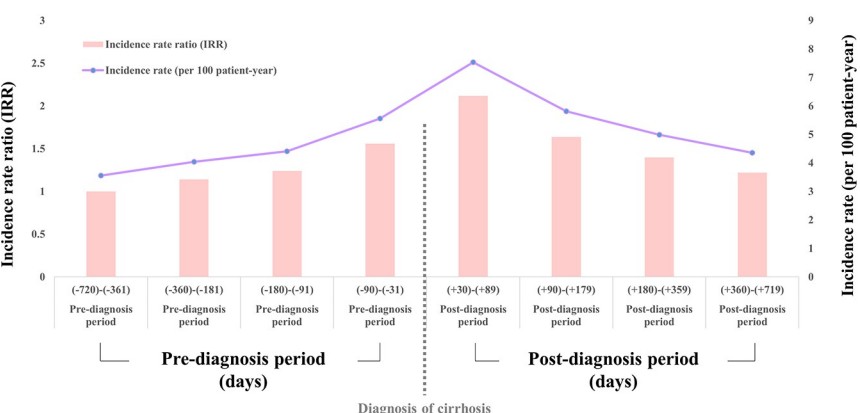

**B**

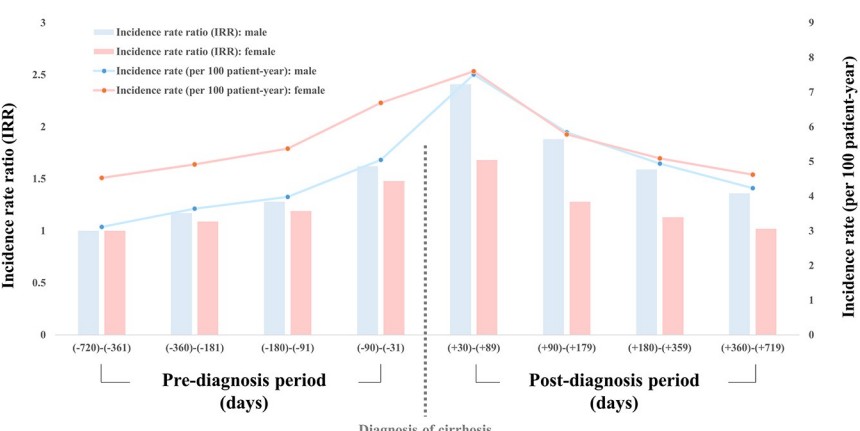

**C**

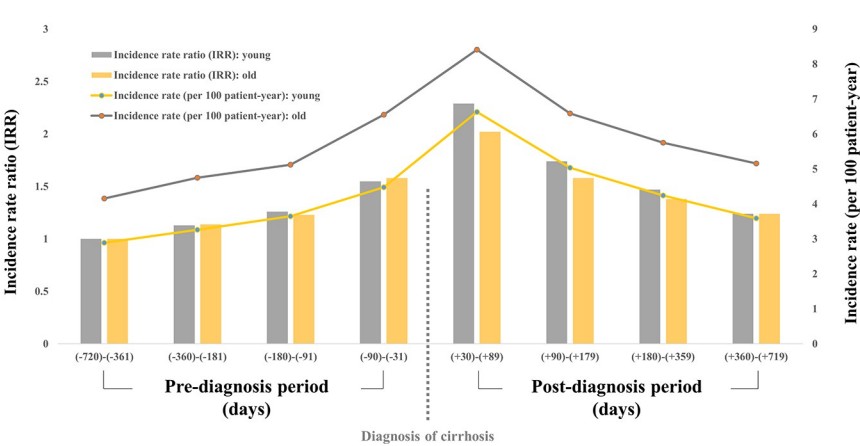

**Fig 2. Incidence rate (IR) and incidence rate ratio (IRR) of depressive disorder in patients with cirrhosis.** (A) Overall, (B) according to sex, (C) according to age group.

was observed in a similar pattern to cancer and chronic lung disease [19]. Our findings suggest that among various possible causes of depression in liver cirrhosis, such as psychological and psychosocial causes, hepatic encephalopathy and drug adverse effects, psychological trauma immediately after diagnosis might be the reason [20]. However, there is a limitation in understanding the exact causal relationship in our study due to the retrospective design.

Patients with cirrhosis are more vulnerable for mental disease such as depression or anxiety. So far, there are not many studies focusing on the mental disease of cirrhotic patients. In a cohort study consisting of 1,021 cirrhotic patients, 15.6% of cirrhotic patients had moderately severe or more depression, and about half of them were diagnosed with anxiety based on the telephone-based survey [21]. In the past, it was reported that the prevalence of depression was high only in the group with poor liver function [22–24], but in a recent study, high rates of MDD and anxiety were reported even in compensated cirrhosis [21]. Although the exact pathophysiology of how liver cirrhosis can increase psychiatric disorders such as depression and anxiety has not been elucidated, but substance alcohol abuse, unhealthy eating habits, and lack of physical exercise might be the reason. In particular, since depression is reported higher in men, being widowed, self-reported poor health, and Hispanic ethnicity groups, screening for depression in these high-risk group is actively required [21].

Moreover, the increase of IRR of depression within first three months of cirrhosis diagnosis was equally observed in all subgroups: male, female, old or young. Although there are some differences in IRR by gender and age, it is a clinically noteworthy point that the diagnosis of liver cirrhosis increases depression in all groups.

Another notable point in our study is that the incidence of MDD is increasing 1 year before compared to the control period of 1 to 2 years before the diagnosis of cirrhosis. Similarly, an increase in MDD before diagnosis of chronic disease has been reported [10]. In fact, before being formally diagnosed with cirrhosis, patients often complain of subjective symptoms such as fatigue, abdominal discomfort, and pruritis due to abnormal liver function, jaundice, or ascites. In addition, cirrhosis is often accompanied by several other chronic diseases. Therefore, we interpreted that the incidence of MDD was already increased due to these symptoms or discomforts, which gradually increased before a formal diagnosis of cirrhosis.

Our study has several limitations. First, because depression was only defined by the claim code for major depressive, depression defined in our study and depression diagnosed in actual clinical settings may differ vastly. Secondly, our study only analyzed the incidence of depression, not its following treatment or progression. In the long run, in order to improve the lives of cirrhotic patients, further research on accompanied depression is essential.

Third, we calculated the incidence only within one group of cirrhosis without comparison with other groups. The most accurate way to see a comparison of incidence rates is a randomized controlled trial (RCT). However, due to the nature of our research topic, it was impossible to conduct RCT, and it was also difficult to conduct a cohort study with different groups due to heterogenicity and comparability. Therefore, we chose the method of comparing the incidence rates before and after the diagnosis of liver cirrhosis in the same patient. Lastly, with the current structure of the study, it is not possible to elucidate the exact role of clinical factors that can influence cirrhosis and depression, such as medically controlling for depression, curing hepatitis C disease, or fibrosis regression in nonalcoholic steatohepatitis. In order to know a more accurate causal relationship, it is very important to accurately obtain clinical information such as medically controlling for depression and curing hepatitis C disease. However, this

study was based on claim data, and detailed data on the etiology of liver cirrhosis (especially alcohol, nonalcoholic fatty liver disease), whether fibrosis has regressed, or whether sustained viral response has obtained after HCV treatment is not available. We think that more well-organized prospective cohort is more appropriate than claim data to study how detailed clinical factors affect the occurrence of depression.

In conclusion, our findings on the unique pattern of depression in cirrhosis have important implications for physicians. Clinicians must pay close attention to screening, preventing and treating depression within the first three months of liver cirrhosis diagnosis. In addition, since the incidence of depression is relatively still high even after three months of diagnosis, a regular checkup of depression, rather than a single visit, is recommended for the mental stability and the consequential long-term satisfaction in cirrhotic patients.

## Acknowledgments

We would like to thank Jieun Lee and Research Factory Publication for editing and reviewing this manuscript for English language.

**Patient consent statement:** Informed consent was waived form the IRB due to the retrospective design.

## Author Contributions

**Conceptualization:** Jeong-Ju Yoo.

**Formal analysis:** Gi Hyeon Seo.

**Methodology:** Gi Hyeon Seo.

**Supervision:** Gi Hyeon Seo.

**Writing – original draft:** Jeong-Ju Yoo.

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
