## [Decision Letter · Decision Letter 0]

11 Oct 2022

PONE-D-22-24164Incidence of major depressive disorder over time in patients with liver cirrhosis: A nationwide population-based study in KoreaPLOS ONE

Dear Dr. Yoo,

Thank you for submitting your manuscript to PLOS ONE. After careful consideration, we feel that it has merit but does not fully meet PLOS ONE’s publication criteria as it currently stands. Therefore, we invite you to submit a revised version of the manuscript that addresses the points raised during the review process.

We look forward to receiving your revised manuscript.

Kind regards,

Sanjiv Mahadeva, MRCP, MD

Academic Editor

PLOS ONE

**Journal Requirements:**

"This study was supported by the Soonchunhyang University Research Fund."

"This study was supported by the Soonchunhyang University Research Fund."

"This study was supported by the Soonchunhyang University Research Fund."

**Additional Editor Comments:**

****The study topic is of interest. However, there are several concerns regarding the study design & methodology. Please refer to the reviewers' comments & address them accordingly.

Reviewers' comments:

Reviewer's Responses to Questions

**Comments to the Author**

1. Is the manuscript technically sound, and do the data support the conclusions?

Reviewer #1: Partly

Reviewer #2: Partly

2. Has the statistical analysis been performed appropriately and rigorously? 

Reviewer #1: No

Reviewer #2: No

3. Have the authors made all data underlying the findings in their manuscript fully available?

Reviewer #1: Yes

Reviewer #2: Yes

4. Is the manuscript presented in an intelligible fashion and written in standard English?

Reviewer #1: Yes

Reviewer #2: Yes

5. Review Comments to the Author

Reviewer #1: This is a study in an area in cirrhosis which has not attracted much attention. Therefore, this manuscript is welcomed. This study is on the incidence of depression based on the timeline relative to the diagosis of cirrhosis, ie. before and after based on reimbursement claims. I have a few comments regarding the manuscript.

MAJOR comments

1. The Abstract lacked details of the data generated from this study (eg. incidence rate of major depression before and/or after diagnosis of cirrhosis). It will be more useful and informative to the reader to include these and other relevant statistics in the Abstract.

2. The Materials and Methods lack a detailed description of the statistical analysis performed for this study. Please highlight the statistical methods on incidence rate and relative risk (or should it be incidence rate ratio?).

3. I am not familiar with the Korean HIRA but if the data is extracted based on claims for reiumbursements, then the number of claims (ie. the numerator) will include patients with existing depression and those with newly diagnosed depression, ie. the prevalence of depression. If the data in this study is based on incidence, then the numerator should include only newly diagnosed depression at each stipulated time frame. It would be useful for the authors to clarify this matter and state this clearly under Materials and Methods.

4. The authors substratified the dataset into smaller groups based on age and gender. I am not versed in the scientific literature of depression but is the incidence (or prevalence) of depression dependent on age and gender ? Please clarify this and provide appropriate references in under Materials and Methods.

5. The authors divided the study period into pre- and post-diagnosis of cirrhosis. Similar to Q4 above on age and gender, what is the relevance of these time intervals (based on number of days before and after) on the diagnosis of depression and provide references if appropriate.

6. A large proportion of the statistical analysis of this study is a comparison of incidence rate of depression after diagnosis of cirrhosis to the incidence rate of depression between 1 to 2 years before diagnosis of cirrhosis. How did the authors compute the relative risk and arrivate at the correspondings statistical p value ? In my opinion, it would be more appropriate to compare incidence rate of different groups by the incidence rate ratio, ie. the IRR. The statistical method for comparison would either be Poisson or negative binomial regression depending on characteristics of the dataset.

MINOR comments

1. In paragraph 2, lines 1-5 of Materials and Methods, the authors described the exclusion criteria for the study. I am interested to know why each exclusion criteria is necessary and provide the apropriate references.

2. In paragraph 2, line 2 of Results, the mean age was expressed as "58.5+13.5 years". This is likely a typographical error and the standard deviation of the mean should be expressed as "58.5±13.5 years".

3. In paragraph 1 of Discussion, the authors described the increasing trend of incidence of depression after the diagnosis of cirrhosis which gradually decreases over time. It is also worthy to comment that in the 1 year preceeding the diagnosis of cirrhosis, the incidence of depression was already increasing compared to 1 to 2 years prior to diagnosis regardless of age and gender, which the authors failed to note. I am curious why this might be the case. Is this interesting observation also seen in other chronic diseases ?

4. In paragraph 2, lines 5 & 6 of Discussion, the authors concluded that "....because of high basal incidence, the relative risks were lower in female and elderly patients......". To substantiate this, suggest to compare the incidence rate of depression at baseline across groups, ie. 3.11 vs 4.53 per 100 patients-years for comparison of male vs female and 2.89 vs 4.16 per 100 patient years for comparison of young vs old and compute the appropriate p value for statistical signifiance of the comparison.

5. Under Discussion, there is little mention on other relevant scientific literature on depression and/or anxiety in cirrhosis or the possible reasons for aggravation of depression in this group of patients. The authors provide a single reference 6 but this is a narrative review and is not specific to cirrhosis. Hernaez R et al. Clin Gastroenterol Hepatol 2022 provided an insight to the risk factors for depression and anxiety in cirrhosis.

6. Similar to the comments on relative risk or more appropriately IRR, Table 1 should reflect the appropriate statistical method of comparison. There is also no mention of the p value in the comparison for the substratification by age and gender.

7. The x-axis of Figure 1A, 1B and 1C is ambiguous. It should be reflected clearly that the numbers on the x-axis refers to days. Please also ensure the right y-axis is labelled appropriately, ie. RR or IRR?

Reviewer #2: 1. The purpose of the article was to examine the incidence of major depressive disorder (MDD) among patients with cirrhosis from the time before and after their diagnosis, using a retrospective analysis of the ICD-10.

2. There was a very interesting study, and I have the following concerns to address: kindly be informed that the author provided details regarding the materials and techniques, including the methodology, flow diaphragm of study enrollment using inclusion and exclusion criteria, as well as statistical analysis used in the article.

3. Population: Due to the article's focus on the incidence of major depressive disorder (MDD) in cirrhotic patients both before and after diagnosis, the precise moment of each disease's diagnosis had a significant impact on the study's findings. How does the author use the ICD-10's first-ever documentation of "cirrhosis" and "depression"? In the methodology, please be more specific.

4. What does "an incorrect mortality code" mean in the context of the materials and procedures on the 48th line?

5. Would the author kindly clarify on how they classified the time before and after the cirrhosis diagnosis using eight periods?

6. Following the result section on lines 78 to 82 of the paragraph as follows: “possibly due to the aggravation of underlying condition before cirrhosis (e.g. chronic hepatitis) that gradually deteriorated patients’ mental health.” Are there any supporting facts or just the author's opinions? Please reconsider the phase change or switch to the discussion part.

7. According to the author mention in the discussion part: "psychological trauma immediately after diagnosis might be the most influential factor." From the design of the retrospective study, there was overemphasis. Kindly ask the author to reconsider.

8. The author mentions the following in the discussion section: “because of the high basal incidence, the relative risks were lower in females and elderly patients than in males and younger patients.” According to another statement, "which even increased after the start of liver cirrhosis," it was difficult for the reader to understand and may have been misunderstood.

9. Additionally, the majority of illnesses that have an impact on mental health or depression are chronic illnesses such chronic liver disease or cancer, including hepatocellular carcinoma. Some cirrhotic patients, particularly those with a Child-Pugh score of A or treatable illnesses like hepatitis C infection, may not experience symptoms of their condition. This article lacked important information.

10. Last but not least, the disease's treatment for both cirrhosis and depression affects the incidence in each stage of the illness, such as medically controlling for depression, curing hepatitis C disease, and fibrosis regression in NASH. Once more, this articel was lacking crucial details.

11. Minor comment: Figure 1's right column contains an incorrect term for the “pre-diagnosis stage” of cirrhosis. Kindly ask the author to reconsider.

6. PLOS authors have the option to publish the peer review history of their article (what does this mean?). If published, this will include your full peer review and any attached files.

Reviewer #1: No

Reviewer #2: No

---

## [Author Response · Author response to Decision Letter 0]

22 Oct 2022

(Please refer the attached word file.)

Responses to the Reviewers’ Comments

19 October, 2022

Dear reviewers and editorial staff of PLOS ONE

We extend our sincere gratitude for your thorough consideration and scrutiny of our manuscript, “Incidence of major depressive disorder over time in patients with liver cirrhosis: A nationwide population-based study in Korea”, control number PONE-D-22-24164. The accurate comments of the reviewers have helped us to better understand the critical issues of this paper. We have revised the manuscript according to the reviewers’ suggestions. We hope that our revised manuscript will be considered and accepted for publication in PLOS ONE. We acknowledge that the scientific and clinical quality of our manuscript was improved by the scrutinizing efforts of the reviewers and editors.

The changes made within the revised manuscript were highlighted (underlined and in blue). Point-by-point responses to the reviewers’ comments are provided below.

Reviewer #1 :

<MAJOR COMMENTS>

1) Reviewer’s comment: The Abstract lacked details of the data generated from this study (eg. incidence rate of major depression before and/or after diagnosis of cirrhosis). It will be more useful and informative to the reader to include these and other relevant statistics in the Abstract.

Author’s response: We appreciate the reviewer’s insightful comment. We added more detailed data to the abstract as the reviewer’s suggestion. 

“There is yet to be a large-scale longitudinal study on the course of depression incidence within the duration of cirrhosis. The aim of this study is to analyze the incidence of depression from before to after diagnosis of cirrhosis over time. Incidence Rate (IR) was defined as the number of newly diagnosed patients with MDD divided by the sum of observation periods by using claims database in Korea. Incidence Rate Ratio (IRR) was defined as the IR in the specific interest period divided by the IR in the control period. The IRs before and after cirrhosis diagnosis were 3.56 and 7.54 per 100 person-year, respectively. The IRR was 2.12 (95% confidence Interval: 2.06-2.18). The IRR of developing depression mildly increased before diagnosis of cirrhosis (-360 days to -181 days, IRR 1.14; -180 days to -90 days, IRR 1.24; -90 days to -31 days, IRR 1.56) and rapidly increased immediately after diagnosis of cirrhosis (+30 days to +89 days, IRR 2.12, 95% confidence interval: 2.06-2.18). The pattern of increasing depression immediately after the diagnosis of cirrhosis was observed equally in all sexes and ages. Thus, clinicians must pay close attention to screening for depression within the first three months of liver cirrhosis diagnosis.”

2) Reviewer’s comment: The Materials and Methods lack a detailed description of the statistical analysis performed for this study. Please highlight the statistical methods on incidence rate and relative risk (or should it be incidence rate ratio?).

Author’s response: We thank the reviewer for enhancing the quality of our manuscript. According to the reviewer's opinion, "relative risk" was changed to "incidence rate ratio" throughout the entire paper to avoid confusion. We also added more detailed description of the statistical analysis as below. 

“Incidence rate (IR) was defined as the number of newly diagnosed patients with MDD divided by the sum of observation periods. Incidence rate ratio (IRR) was defined as the IR in the specific interest period divided by the IR in the control period. The 95% confidence interval (CI) and p-value were calculated using a Poisson approximation. In addition, we conducted stratified analysis according to age and sex considering the past results that the prevalence of depression differs according to age and sex.” 

3) Reviewer’s comment: I am not familiar with the Korean HIRA but if the data is extracted based on claims for reiumbursements, then the number of claims (ie. the numerator) will include patients with existing depression and those with newly diagnosed depression, ie. the prevalence of depression. If the data in this study is based on incidence, then the numerator should include only newly diagnosed depression at each stipulated time frame. It would be useful for the authors to clarify this matter and state this clearly under Materials and Methods.

Author’s response: We strongly agree with the reviewer’s opinion. As the reviewer said, we only included the newly diagnosed depression. We excluded patients diagnosed with MDD 720 days prior to the diagnosis of cirrhosis in January 2007. We added more clearly to Method as follows. 

“First-ever documentation of cirrhosis was defined as a new patient with no treatment for liver cirrhosis for at least 3 years (January 2007 – December 2009 wash-out), and major depressive disorder (MDD) was defined a patient who had no treatment for MDD for at least 1 year (between January 2007 and 720 days prior to the diagnosis of cirrhosis) as a new patient.”

4) Reviewer’s comment: The authors substratified the dataset into smaller groups based on age and gender. I am not versed in the scientific literature of depression but is the incidence (or prevalence) of depression dependent on age and gender ? Please clarify this and provide appropriate references in under Materials and Methods.

Author’s response: We are grateful for the reviewer’s thoughtful comment. We additionally added the references for stratification according to age and gender in the method.

“In addition, we conducted stratified analysis according to age and sex considering the past results that the prevalence of depression differs according to age and sex,[1-6]”

5) Reviewer’s comment: The authors divided the study period into pre- and post-diagnosis of cirrhosis. Similar to Q4 above on age and gender, what is the relevance of these time intervals (based on number of days before and after) on the diagnosis of depression and provide references if appropriate.

Author’s response: Thank you for your thoughtful insight. We added the evidence of time intervals used in our study as follows: 

“The time interval was determined considering on the length of depressive episode, average duration of evaluation of improvement in depression, and the impact of acute event such as diagnosis of liver cirrhosis.[7-9] Considering that cirrhosis and MDD are chronic diseases, the baseline was set to 1 year 2 years ago before diagnosis. Assuming that the patient's condition can change more frequently around the onset of cirrhosis, the analysis was conducted at shorter intervals of 180 days, 90 days, and 60 days closer to the date of diagnosis of cirrhosis. Based on this information, the duration of liver cirrhosis was divided into a total of eight periods.”

6) Reviewer’s comment: A large proportion of the statistical analysis of this study is a comparison of incidence rate of depression after diagnosis of cirrhosis to the incidence rate of depression between 1 to 2 years before diagnosis of cirrhosis. How did the authors compute the relative risk and arrivate at the correspondings statistical p value ? In my opinion, it would be more appropriate to compare incidence rate of different groups by the incidence rate ratio, ie. the IRR. The statistical method for comparison would either be Poisson or negative binomial regression depending on characteristics of the dataset.

Author’s response: We strongly agree with the reviewer’s opinion. As you said, comparing incidence rates in different populations groups is a more accurate and better method. However, there are few previous studies on the incidence of MDD in cirrhosis patients, and the results on the incidence of MDD in patients with cirrhosis compared to the heathy population or other chronic diseases are still lacking. Therefore, we judged that it was difficult to select a control group with a similar risk of MDD other than liver cirrhosis, so we used a method of comparing the incidence rates before and after the diagnosis of liver cirrhosis in the same patient. We added this limitation to the Discussion section as follows. 

“Third, we calculated the incidence only within one group of cirrhosis without comparison with other groups. The most accurate way to see a comparison of incidence rates is a randomized controlled trial (RCT). However, due to the nature of our research topic, it was impossible to conduct RCT, and it was also difficult to conduct a cohort study with different groups due to heterogenicity and comparability. Therefore, we chose the method of comparing the incidence rates before and after the diagnosis of liver cirrhosis in the same patient.”

As for the statistical method, the 95% confidence interval and p-value were calculated using a Poisson approximation. We added more detailed description of the statistical analysis as below. 

“Incidence rate (IR) was defined as the number of newly diagnosed patients with MDD divided by the sum of observation periods. Incidence rate ratio (IRR) was defined as the IR in the specific interest period divided by the IR in the control period. The 95% confidence interval (CI) and p-value were calculated using a Poisson approximation.”

<MINOR COMMENTS>

1) Reviewer’s comment: In paragraph 2, lines 1-5 of Materials and Methods, the authors described the exclusion criteria for the study. I am interested to know why each exclusion criteria is necessary and provide the apropriate references.

Author’s response: We appreciate the reviewer’s insightful comment. We added the reason and references of each exclusion criteria as the reviewer’s recommendation. 

“Out of 787,505 adult patients treated for liver cirrhosis from 2010 to 2018, patients groups including those aged 18 years or younger, or 89 years or older, considering the average life expectancy in Korea (n=7,953); those diagnosed with cirrhosis from 2007 to 2009, to include only newly diagnosed cirrhosis patients during the study period; (n=175,608); those diagnosed with depression 720 days prior to the diagnosis of cirrhosis as a wash-out period to include only newly diagnosed MDD[10] (n = 93,164) and those with an incorrect mortality code such as the date of death precedes the date of diagnosis due to administrative error; (n = 43) were excluded.”

2) Reviewer’s comment: In paragraph 2, line 2 of Results, the mean age was expressed as "58.5+13.5 years". This is likely a typographical error and the standard deviation of the mean should be expressed as "58.5±13.5 years".

Author’s response: We appreciate the reviewer’s insightful comment. We corrected the typographical error. 

3) Reviewer’s comment: In paragraph 1 of Discussion, the authors described the increasing trend of incidence of depression after the diagnosis of cirrhosis which gradually decreases over time. It is also worthy to comment that in the 1 year preceeding the diagnosis of cirrhosis, the incidence of depression was already increasing compared to 1 to 2 years prior to diagnosis regardless of age and gender, which the authors failed to note. I am curious why this might be the case. Is this interesting observation also seen in other chronic diseases ? 

Author’s response: We thank the reviewer for enhancing the quality of our manuscript. In women who underwent assisted reproduction technology (ART) for infertility, there was a report that depression increased 2-3 years before ART (Please refer the below figure). 

Before being formally diagnosed with cirrhosis, patients often complain of subjective symptoms such as fatigue, abdominal discomfort, and pruritis due to abnormal liver function, jaundice, or ascites. In addition, cirrhosis is often accompanied by several other chronic diseases. Therefore, we interpreted that the incidence of MDD was already increased due to these symptoms or discomforts, which gradually increased before a formal diagnosis of cirrhosis. This part was additionally mentioned and reinforced in the discussion as follows. 

“Another notable point in our study is that the incidence of MDD is increasing 1-2 years before the diagnosis of cirrhosis. Similarly, an increase in MDD before diagnosis of chronic disease has been reported.[9] In fact, before being formally diagnosed with cirrhosis, patients often complain of subjective symptoms such as fatigue, abdominal discomfort, and pruritis due to abnormal liver function, jaundice, or ascites. In addition, cirrhosis is often accompanied by several other chronic diseases. Therefore, we interpreted that the incidence of MDD was already increased due to these symptoms or discomforts, which gradually increased before a formal diagnosis of cirrhosis.”

4) Reviewer’s comment: In paragraph 2, lines 5 & 6 of Discussion, the authors concluded that "....because of high basal incidence, the relative risks were lower in female and elderly patients......". To substantiate this, suggest to compare the incidence rate of depression at baseline across groups, ie. 3.11 vs 4.53 per 100 patients-years for comparison of male vs female and 2.89 vs 4.16 per 100 patient years for comparison of young vs old and compute the appropriate p value for statistical signifiance of the comparison.

Author’s response: We are grateful for the reviewer’s thoughtful comment. We added appropriate p value for Table 1 for comparison of young vs. old or male vs. female in Table 1. 

However, the purpose of the stratification analysis was to emphasize the fact that MDD increased immediately after the diagnosis of cirrhosis in all subgroups. Therefore, the comparison with respect to gender and age is different from the original purpose of this study. It seems that the description or emphasis of the text was misrepresented in the original text, so it was rewritten to emphasize the original purpose of the study as follows.

“Moreover, the increase of IRR of depression within first three months of cirrhosis diagnosis was equally observed in all subgroups: male, female, old or young. Although there are some differences in IRR by gender and age, it is a clinically noteworthy point that the diagnosis of liver cirrhosis increases depression in all groups”

5) Reviewer’s comment: Under Discussion, there is little mention on other relevant scientific literature on depression and/or anxiety in cirrhosis or the possible reasons for aggravation of depression in this group of patients. The authors provide a single reference 6 but this is a narrative review and is not specific to cirrhosis. Hernaez R et al. Clin Gastroenterol Hepatol 2022 provided an insight to the risk factors for depression and anxiety in cirrhosis.

Author’s response: Thank you for your thoughtful insight. We added the possible mechanism for increasing MDD in cirrhotic patients, and added the reference as the reviewer’s suggestion. Thank you so much for recommending this appropriate reference. 

“Patients with cirrhosis are more vulnerable for mental disease such as depression or anxiety. So far, there are not many studies focusing on the mental disease of cirrhotic patients. In a cohort study consisting of 1,021 cirrhotic patients, 15.6% of cirrhotic patients had moderately severe or more depression, and about half of them were diagnosed with anxiety based on the telephone-based survey.[11] In the past, it was reported that the prevalence of depression was high only in the group with poor liver function[12-14], but in a recent study, high rates of MDD and anxiety were reported even in compensated cirrhosis.[11] Although the exact pathophysiology of liver cirrhosis increases psychiatric disorders such as depression and anxiety, the exact pathophysiology has not been elucidated, but substance alcohol abuse, unhealthy eating habits, and lack of physical exercise might be the reason. In particular, since depression is reported higher in men, being widowed, self-reported poor health, and Hispanic ethnicity groups, screening for depression in these high-risk group is actively required.[11]”

6) Reviewer’s comment: Similar to the comments on relative risk or more appropriately IRR, Table 1 should reflect the appropriate statistical method of comparison. There is also no mention of the p value in the comparison for the substratification by age and gender.

Author’s response: Thank you for your thoughtful insight. According to the reviewer's opinion, "relative risk" was changed to "incidence rate ratio" throughout the entire paper to avoid confusion. About the statistical method, we added more detailed description of the statistical analysis as below. 

“The 95% confidence interval and p-value were calculated using a Poisson approximation.”

In Table 1, we added p value in the comparison for the substratification by age and gender as follows. 

7) Reviewer’s comment: The x-axis of Figure 1A, 1B and 1C is ambiguous. It should be reflected clearly that the numbers on the x-axis refers to days. Please also ensure the right y-axis is labelled appropriately, ie. RR or IRR?

Author’s response: Thank you for your thoughtful insight. We added the labelling of “days” to X-axis. Also, we have dual y-axis (right and left), the left one is “incidence rate ratio (IRR)” and the right one is “incidence rate (IR)”. The Figure was revised as follows. 

Reviewer #2 :

<MAJOR COMMENTS>

1) Reviewer’s comment: The purpose of the article was to examine the incidence of major depressive disorder (MDD) among patients with cirrhosis from the time before and after their diagnosis, using a retrospective analysis of the ICD-10. There was a very interesting study, and I have the following concerns to address: kindly be informed that the author provided details regarding the materials and techniques, including the methodology, flow diaphragm of study enrollment using inclusion and exclusion criteria, as well as statistical analysis used in the article. 

Author’s response: We appreciate the reviewer’s insightful comment. We added the flow diaphragm of study as follows. 

About the statistical method, we added more detailed description of the statistical analysis as below. 

“Incidence rate (IR) was defined as the number of newly diagnosed patients with MDD divided by the sum of observation periods. Incidence rate ratio (IRR) was defined as the IR in the specific interest period divided by the IR in the control period. The 95% confidence interval (CI) and p-value were calculated using a Poisson approximation.”

2) Reviewer’s comment: Population: Due to the article's focus on the incidence of major depressive disorder (MDD) in cirrhotic patients both before and after diagnosis, the precise moment of each disease's diagnosis had a significant impact on the study's findings. How does the author use the ICD-10's first-ever documentation of "cirrhosis" and "depression"? In the methodology, please be more specific.

Author’s response: Thank you for your thoughtful insight. First-ever documentation of cirrhosis was defined as a new patient with no treatment for liver cirrhosis for at least 3 years (July 2009 – December 2009 wash-out), and major depressive disorder (MDD) was defined a patient who had no treatment for MDD for at least 1 year (720 days before diagnosis of cirrhosis after January 2007) as a new patient. We added this information to the Method section. 

“First-ever documentation of cirrhosis was defined as a new patient with no treatment for liver cirrhosis for at least 3 years (January 2007 – December 2009 wash-out), and major depressive disorder (MDD) was defined a patient who had no treatment for MDD for at least 1 year (between January 2007 and 720 days prior to the diagnosis of cirrhosis) as a new patient.”

3) Reviewer’s comment: What does "an incorrect mortality code" mean in the context of the materials and procedures on the 48th line?

Author’s response: We are grateful again for your thoughtful comment. We describe the method in more detail as follows.

“those with an incorrect mortality code such as the date of death precedes the date of diagnosis due to administrative error; (n = 43) were excluded.”

4) Reviewer’s comment: Would the author kindly clarify on how they classified the time before and after the cirrhosis diagnosis using eight periods?

Author’s response: Thank you for your thoughtful insight. We added the evidence of time intervals used in our study as follows: 

“The time interval was determined considering on the length of depressive episode, average duration of evaluation of improvement in depression, and the impact of acute event such as diagnosis of liver cirrhosis.[7-9] Considering that cirrhosis and MDD are chronic diseases, the baseline was set to 1 year 2 years ago before diagnosis. Assuming that the patient's condition can change more frequently around the onset of cirrhosis, the analysis was conducted at shorter intervals of 180 days, 90 days, and 60 days closer to the date of diagnosis of cirrhosis. Based on this information, the duration of liver cirrhosis was divided into a total of eight periods.”

5) Reviewer’s comment: Following the result section on lines 78 to 82 of the paragraph as follows: “possibly due to the aggravation of underlying condition before cirrhosis (e.g. chronic hepatitis) that gradually deteriorated patients’ mental health.” Are there any supporting facts or just the author's opinions? Please reconsider the phase change or switch to the discussion part.

Author’s response: We are grateful again for your thoughtful comment. This was our hypothesis, and we moved this part to Discussion section as follows. 

“Another notable point in our study is that the incidence of MDD is increasing 1-2 years before the diagnosis of cirrhosis. Similarly, an increase in MDD before diagnosis of chronic disease has been reported.[9] In fact, before being formally diagnosed with cirrhosis, patients often complain of subjective symptoms such as fatigue, abdominal discomfort, and pruritis due to abnormal liver function, jaundice, or ascites. In addition, cirrhosis is often accompanied by several other chronic diseases. Therefore, we interpreted that the incidence of MDD was already increased due to these symptoms or discomforts, which gradually increased before a formal diagnosis of cirrhosis.”

6) Reviewer’s comment: According to the author mention in the discussion part: "psychological trauma immediately after diagnosis might be the most influential factor." From the design of the retrospective study, there was overemphasis. Kindly ask the author to reconsider.

Author’s response: We are grateful again for your thoughtful comment. The tone of the sentence has been slightly relaxed as follows.

“Our findings suggest that among various possible causes of depression in liver cirrhosis, such as psychological and psychosocial causes, hepatic encephalopathy and drug adverse effects, psychological trauma immediately after diagnosis might be the reason. However, there is a limitation in understanding the exact causal relationship in our study due to the retrospective design.”

7) Reviewer’s comment: The author mentions the following in the discussion section: “because of the high basal incidence, the relative risks were lower in females and elderly patients than in males and younger patients.” According to another statement, "which even increased after the start of liver cirrhosis," it was difficult for the reader to understand and may have been misunderstood.

Author’s response: We appreciate the reviewer’s insightful comment. The purpose of the stratification analysis was to emphasize the fact that MDD increased immediately after the diagnosis of cirrhosis in all subgroups. Therefore, the comparison with respect to gender and age is different from the original purpose of this study. It seems that the description or emphasis of the text was misrepresented in the original text, so it was rewritten to emphasize the original purpose of the study as follows.

“Moreover, the increase of IRR of depression within first three months of cirrhosis diagnosis was equally observed in all subgroups: male, female, old or young. Although there are some differences in IRR by gender and age, it is a clinically noteworthy point that the diagnosis of liver cirrhosis increases depression in all groups”

8) Reviewer’s comment: Additionally, the majority of illnesses that have an impact on mental health or depression are chronic illnesses such chronic liver disease or cancer, including hepatocellular carcinoma. Some cirrhotic patients, particularly those with a Child-Pugh score of A or treatable illnesses like hepatitis C infection, may not experience symptoms of their condition. This article lacked important information. 

Author’s response: We are grateful again for your thoughtful comment. We wanted to emphasize that in the past, it was reported that the prevalence of depression was high only in the group with poor liver function, but in a recent study, high rates of MDD and anxiety were reported even in compensated cirrhosis. We further described for the possible mechanism for increasing MDD in cirrhotic patients in Discussion section as below. 

“Patients with cirrhosis are more vulnerable for mental disease such as depression or anxiety. So far, there are not many studies focusing on the mental disease of cirrhotic patients. In a cohort study consisting of 1,021 cirrhotic patients, 15.6% of cirrhotic patients had moderately severe or more depression, and about half of them were diagnosed with anxiety based on the telephone-based survey.[11] In the past, it was reported that the prevalence of depression was high only in the group with poor liver function[12-14], but in a recent study, high rates of MDD and anxiety were reported even in compensated cirrhosis.[11] Although the exact pathophysiology of liver cirrhosis increases psychiatric disorders such as depression and anxiety, the exact pathophysiology has not been elucidated, but substance alcohol abuse, unhealthy eating habits, and lack of physical exercise might be the reason. In particular, since depression is reported higher in men, being widowed, self-reported poor health, and Hispanic ethnicity groups, screening for depression in these high-risk group is actively required.[11]”

9) Reviewer’s comment: Last but not least, the disease's treatment for both cirrhosis and depression affects the incidence in each stage of the illness, such as medically controlling for depression, curing hepatitis C disease, and fibrosis regression in NASH. Once more, this articel was lacking crucial details.

Author’s response: We fully agree with the reviewer’s opinion. In order to know a more accurate causal relationship, it is very important to accurately obtain clinical information such as medically controlling for depression and curing hepatitis C disease. However, this study was based on claim data, and detailed data on the etiology of liver cirrhosis (especially alcohol, nonalcoholic fatty liver disease), whether fibrosis has regressed, or whether sustained viral response has obtained after HCV treatment is not available. We think that more well-organized prospective cohort is more appropriate than claim data to study how detailed clinical factors affect the occurrence of depression. Thus, we are planning a prospective cohort study on the same topic as a follow-up study. In the present study, the reviewer's opinion is clinically very important and we fully agree, and the limitation is added as follows. 

“Lastly, with the current structure of the study, it is not possible to elucidate the exact role of clinical factors that can influence cirrhosis and depression, such as medically controlling for depression, curing hepatitis C disease, or fibrosis regression in nonalcoholic steatohepatitis. In order to know a more accurate causal relationship, it is very important to accurately obtain clinical information such as medically controlling for depression and curing hepatitis C disease. However, this study was based on claim data, and detailed data on the etiology of liver cirrhosis (especially alcohol, nonalcoholic fatty liver disease), whether fibrosis has regressed, or whether sustained viral response has obtained after HCV treatment is not available. We think that more well-organized prospective cohort is more appropriate than claim data to study how detailed clinical factors affect the occurrence of depression.”

<MINOR COMMENTS>

1) Reviewer’s comment: Figure 1's right column contains an incorrect term for the “pre-diagnosis stage” of cirrhosis. Kindly ask the author to reconsider.

Author’s response: Thank you for your thoughtful insight. We corrected the right column to the right term. Also, we added the labelling of “days” to X-axis. The Figure was revised as follows.

---

## [Decision Letter · Decision Letter 1]

10 Nov 2022

PONE-D-22-24164R1Incidence of major depressive disorder over time in patients with liver cirrhosis: A nationwide population-based study in KoreaPLOS ONE

Dear Dr. Yoo,

Thank you for submitting your manuscript to PLOS ONE. After careful consideration, we feel that it has merit but does not fully meet PLOS ONE’s publication criteria as it currently stands. Therefore, we invite you to submit a revised version of the manuscript that addresses the points raised during the review process.

We look forward to receiving your revised manuscript.

Kind regards,

Sanjiv Mahadeva, MRCP, MD

Academic Editor

PLOS ONE

Journal Requirements:

Additional Editor Comments:

The revised submission is much improved. Please refer to Reviewer #1's comments & address them accordingly. 

Reviewers' comments:

Reviewer's Responses to Questions

**Comments to the Author**

1. If the authors have adequately addressed your comments raised in a previous round of review and you feel that this manuscript is now acceptable for publication, you may indicate that here to bypass the “Comments to the Author” section, enter your conflict of interest statement in the “Confidential to Editor” section, and submit your "Accept" recommendation.

Reviewer #1: All comments have been addressed

Reviewer #2: All comments have been addressed

2. Is the manuscript technically sound, and do the data support the conclusions?

Reviewer #1: Yes

Reviewer #2: Yes

3. Has the statistical analysis been performed appropriately and rigorously? 

Reviewer #1: Yes

Reviewer #2: Yes

4. Have the authors made all data underlying the findings in their manuscript fully available?

Reviewer #1: No

Reviewer #2: Yes

5. Is the manuscript presented in an intelligible fashion and written in standard English?

Reviewer #1: Yes

Reviewer #2: Yes

6. Review Comments to the Author

Reviewer #1: Thank you for addressing the issues raised in the previous manuscript, especially clarifying the main aim of the research. I have additional minor comments only:

1. Under Abstract and Results, the IRR was reported but no p value was given. Kindly include the p values for the relevant IRRs in the Abstract and Results.

2. Under Abstract, it will also be useful to the reader for authors to define the control period as 1 to 2 years before diagnosis of cirrhosis.

3. Lastly, this is pertaining to the Reviewer #1, Minor Comments 3 on rising incidence rate of depression 1 year before diagnosis compared to control period of 1 to 2 years before diagnosis of cirrhosis. Could the authors consider amending lines 1 to 2 of paragraph 4 under Discussion as "Another notable point in our study is that the incidence of MDD is increasing 1 year before compared to the control period of 1 to 2 years before the diagnosis of cirrhosis".

Reviewer #2: Thank you for giving me the opportunity to again review this interesting paper. Even though the paper still had some limitations, the authors stated and provided these limitations in the discussion section. As well, the authors emphasized key points and responded all of my comments nicely.

7. PLOS authors have the option to publish the peer review history of their article (what does this mean?). If published, this will include your full peer review and any attached files.

Reviewer #1: No

Reviewer #2: No

---

## [Author Response · Author response to Decision Letter 1]

11 Nov 2022

Responses to the Reviewers’ Comments

11 November, 2022

Dear reviewers and editorial staff of PLOS ONE

We extend our sincere gratitude for your thorough consideration and scrutiny of our manuscript, “Incidence of major depressive disorder over time in patients with liver cirrhosis: A nationwide population-based study in Korea”, control number PONE-D-22-24164R1. The accurate comments of the reviewers have helped us to better understand the critical issues of this paper. We have revised the manuscript according to the reviewers’ suggestions. We hope that our revised manuscript will be considered and accepted for publication in PLOS ONE. We acknowledge that the scientific and clinical quality of our manuscript was improved by the scrutinizing efforts of the reviewers and editors.

The changes made within the revised manuscript were highlighted (underlined and in blue). Point-by-point responses to the reviewers’ comments are provided below.

Reviewer #1 :

<MAJOR COMMENTS>

1) Reviewer’s comment: Under Abstract and Results, the IRR was reported but no p value was given. Kindly include the p values for the relevant IRRs in the Abstract and Results.

Author’s response: We appreciate the reviewer’s insightful comment. We added p values in the abstract and result section. 

2) Reviewer’s comment: Under Abstract, it will also be useful to the reader for authors to define the control period as 1 to 2 years before diagnosis of cirrhosis.

Author’s response: We thank the reviewer for enhancing the quality of our manuscript. According to the reviewer's opinion, definition of control period was added to the abstract. 

3) Reviewer’s comment: Lastly, this is pertaining to the Reviewer #1, Minor Comments 3 on rising incidence rate of depression 1 year before diagnosis compared to control period of 1 to 2 years before diagnosis of cirrhosis. Could the authors consider amending lines 1 to 2 of paragraph 4 under Discussion as "Another notable point in our study is that the incidence of MDD is increasing 1 year before compared to the control period of 1 to 2 years before the diagnosis of cirrhosis".

Author’s response: We strongly agree with the reviewer’s opinion. We modified the sentence as the reviewer’s recommendation.

---

## [Editor Report · Decision Letter 2]

24 Nov 2022

Incidence of major depressive disorder over time in patients with liver cirrhosis: A nationwide population-based study in Korea

PONE-D-22-24164R2

Dear Dr. Yoo,

We’re pleased to inform you that your manuscript has been judged scientifically suitable for publication and will be formally accepted for publication once it meets all outstanding technical requirements.

Kind regards,

Sanjiv Mahadeva, MRCP, MD

Academic Editor

PLOS ONE

Additional Editor Comments (optional):

Very minor comments from the previous review - all have been addressed

---

## [Editor Report · Acceptance letter]

1 Dec 2022

PONE-D-22-24164R2 

Incidence of major depressive disorder over time in patients with liver cirrhosis: A nationwide population-based study in Korea 

Dear Dr. Yoo:

I'm pleased to inform you that your manuscript has been deemed suitable for publication in PLOS ONE. Congratulations! Your manuscript is now with our production department. 

Kind regards, 

on behalf of

Prof Sanjiv Mahadeva 

Academic Editor

PLOS ONE